# Food Processing and Nutrition Strategies for Improving the Health of Elderly People with Dysphagia: A Review of Recent Developments

**DOI:** 10.3390/foods13020215

**Published:** 2024-01-10

**Authors:** Ting Liu, Jianheng Zheng, Jun Du, Gengsheng He

**Affiliations:** 1Key Laboratory of Public Health Safety of the Ministry of Education, School of Public Health, Fudan University, Shanghai 200032, China; 22111020008@m.fudan.edu.cn; 2Nutrilite Health Institute, Shanghai 200032, China; jennie.zheng@amway.com (J.Z.); eric.du@amway.com (J.D.)

**Keywords:** dysphagia, food texture, swallowing, food processing, thickeners, 3D printing

## Abstract

Dysphagia, or swallowing difficulty, is a common morbidity affecting 10% to 33% of the elderly population. Individuals with dysphagia can experience appetite, reduction, weight loss, and malnutrition as well as even aspiration, dehydration, and long-term healthcare issues. However, current therapies to treat dysphagia can routinely cause discomfort and pain to patients. To prevent these risks, a non-traumatic and effective treatment of diet modification for safe chewing and swallowing is urgently needed for the elderly. This review mainly summarizes the chewing and swallowing changes in the elderly, as well as important risk factors and potential consequences of dysphagia. In addition, three texture-modified food processing strategies to prepare special foods for the aged, as well as the current statuses and future trends of such foods, are discussed. Nonthermal food technologies, gelation, and 3D printing techniques have been developed to prepare soft, moist, and palatable texture-modified foods for chewing and swallowing safety in elderly individuals. In addition, flavor enhancement and nutrition enrichment are also considered to compensate for the loss of sensory experience and nutrients. Given the trend of population aging, multidisciplinary cooperation for dysphagia management should be a top priority.

## 1. Introduction

Around the world, populations are aging at a fast pace. According to the report from the United Nations (the 2022 revision), the share of people aged 65 years or above is expected to grow from 10% in 2022 to 16% in 2050 and the number of older persons (65 years or over) is projected to be more than twice the number of children under age 5 by 2050, reaching 1.5 billion [1]. Consequently, more elderly-oriented products and services need to be provided to assist older people. Eating ability is a fundamental factor in improving health and wellbeing in old age [2]. However, because of poor oral health, sensory impairment, loss of mobility caused by aging or diseases, and even socio-cultural changes, the elderly often cannot intake enough food, making them vulnerable to malnutrition and immunity senescence. Swallowing disorder, also known as dysphagia, is a major health threat to the elderly and presents with symptoms of aspiration, residual or excessive throat clearing, cough, hoarseness, atypical breathing, and repeated swallowing [3]. Dysphagia is classified as either an oropharyngeal (oral) or esophageal (non-oral) condition based on the location of the bolus obstruction [4]. The former has been shown to be more prevalent and serious than the latter in the elderly. In this review, we mainly focus on oral dysphagia and refer to it simply as dysphagia.

In general, dysphagia can occur at any age, though the risk is higher in those older than 65. The prevalence is 13% in the 65–70 age group, 16% in the 70–79 age group, and around 33% in the ≥80 years group [5,6]. Health status is also a prominent risk factor for dysphagia. The data indicate that dysphagia affects about 68% of the elderly population living in nursing homes, 30% of those who have been hospitalized, and 13% to 38% of those living independently [7]. Up to 64% of stroke patients, 93% of dementia patients, and 82% of Parkinson’s disease patients suffer from difficulty in swallowing [4,8]. Among the many consequences derived from dysphagia, nutritional deficiencies, reduced physical fitness and body mass, aspiration pneumonia, and dehydration may occur in these patients. Clinically, stimulation methods and tube feeding are most often used for dysphagic patients but these methods can easily cause discomfort and pain to patients, as well as reduce the quality of life for this population.

Diet modification is a non-traumatic and promising treatment for swallowing disorders. At present, this strategy is mainly accepted by hospitals and nursing homes to meet the special demands of old people with swallowing difficulties by using thickeners or crushing solids into purees. However, this process requires careful consideration and consultation with nutritionists, physicians, or pharmacists, thus limiting its wide promotion, especially among the elderly living at home. Therefore, the food industry must facilitate the healthy aging of this population by applying appropriate technologies. In this review, we first describe the chewing and swallowing changes in elderly people and introduce several important risk factors for, potential consequences of, and suitable management of dysphagia. Then, the dietary characteristics for dysphagia are briefly described based on clinical compensatory management. As a bridge, we summarize the mechanisms and applications of gelation, non-thermal technologies, and 3D printing technologies for developing special food targeted to elderly dysphagic patients. In addition, the strategies for flavor and nutrition enrichment and the status of foods for elderly dysphagic patients in different countries are discussed. Finally, several perspectives for further studies are presented. It is hoped that this review will shed more light on the understanding of dietary requirements for the elderly with dysphagia at the multidisciplinary level and encourage more studies on the topic in order to provide comprehensive solutions regarding aged diets.

## 2. Dysphagia in the Elderly: Causes, Consequences, Assessment, and Management

### 2.1. Causes and Health Consequences of Dysphagia

Generally, oral food processing can be divided into the following four stages: transportation, mastication, bolus formation, and swallowing preparation [5]. Figure 1A describes the whole process with a graphic flowchart. In the initial stage, the food is ingested into the oral cavity through diet tools. When the presence of food is detected by the sensory system, the tongue and palate function together to gradually transport them to the gum ridges. Next, mastication starts with the first bite of solid or semi-solid food and then these foods begin to be cut by the incisors or the canine teeth and then chewed and sheared by the molar teeth in order to reduce food piece sizes [9]. During this stage, saliva produced by the parotid, submandibular, sublingual, and mini glands hydrolyze carbohydrates, dissolve flavor substances, and enable the brain to produce flavor perception. The food granules are lubricated by saliva to form a cohesive bolus to make swallowing easy and smooth [10]. While being chewed, the bolus is moved to the back of the buccal cavity through pressure from the tongue and the oral jaw. Finally, when the bolus possesses appropriate rheological properties, swallowing is triggered through sensory feedback. This swallowing reflex lasts approximately one second [11]. During the swallowing process, the soft palate in the mouth firstly moves upward so that the nasal cavity is covered to avoid food reflux. At the same time, the epiglottis moves downward in order to close the airway before the food is transported to this position, which can prevent food residues from entering the airway and causing diseases such as aspiration pneumonia [10]. While the pharyngeal diaphragm muscle contracts, the food bolus moves into the proximal esophagus [12] (Figure 1A). In fact, the respiratory and feeding tracts are two completely different passages but at the pharynx, both passages are shared and therefore need to be well coordinated during feeding. Then, the food bolus is gradually pushed into the stomach through contractile movements of the esophageal wall, which may take 8 to 20 s [11].

However, physiological dysfunction linked to aging is common in the aged, a group that is prone to the development of dysphagia. For instance, many older adults have a greatly reduced occlusal capacity and with increasing age, the density of oral–facial muscles in the elderly significantly decreases, which may lead to a remarkable reduction in the contraction of oral muscles during chewing and swallowing and an inability to coordinate movements well, resulting in dysphagia and related disorders [6,13]. Furthermore, the salivary secretion capacity decreases at older ages, affecting the flow-ability of food to form safe and easy-to-swallow boluses [14]. Other typical changes include a prolonged oral phase, decreased swallow volume, delayed closure of the larynx, degenerative cervical spine, reduced compensatory capacity of the brain [6], and reduced dental status [15]. Several important risk factors associated with dysphagia in the older population are demonstrated in Figure 1B. In addition to this, the mixture of aging and related disorders, including neurological diseases [16], sarcopenia [15], oropharynx lesions [6], and respiratory diseases [17] is emerging as a significant healthcare concern, as treatments and medicines targeting such diseases could result in decreased oral ability, taste sensitivity, and xerostomia (mouth dryness), which in turn triggers swallowing difficulty and pain during feeding [18].

By reason of weakened physiological function and diseases, the elderly cannot intake foods properly, which in turn leads to malnutrition, dehydration, weight loss, and even aspiration pneumonia [16,19,20]. Of these, malnutrition was found in 17–20% of elderly people living alone with dysphagia, compared to 37–67% of hospitalized seniors. Almost 50% patients with dysphagia in nursing homes were reported to have aspiration pneumonia within a year of admission [21]. Additionally, these complications can also damage the mental health of senior citizens by, for example, inducing fear, embarrassment, depression, and frustration, which may prolong intensive healthcare, cause adverse prognosis, and even increase the case fatality rate of patients [6,22]. Dysphagia has multiple impacts on the health and life quality of older adults and further details are displayed in Figure 2.

### 2.2. Assessment and Management of Dysphagia

Coughing and choking are common symptoms of dysphagia [23]. Yet, due to a lack of awareness, swallowing disorder is greatly underrecognized and, indeed, many patients may incorrectly assume it to be normal aging. Consequently, careful questionnaires and physical examinations may provide clues for the assessment of dysphagia. For example, the 10-item eating assessment tool (EAT-10) [24], the Sydney swallowing questionnaire (SSQ) [8], the water swallowing test (WST) [25], and the volume viscosity swallow test (V-VST) [26] are commonly used by healthcare personnel for dysphagia evaluation. In addition, two instrumental analysis methods, flexible endoscopic evaluation of swallowing (FEES, a limited invasive but non-radiation evaluation of dysphagia with pharyngeal stage) and video-fluoroscopic swallow study (VFSS, a radiation assessment of dysphagia with oral, pharyngeal, and esophageal stages), are available for the clinical diagnosis of swallowing dysfunction [16].

Once diagnosed, effective dysphagia management can reduce the morbidity of aspiration and malnutrition as well as its consequences, thus recovering swallowing functions and improving life quality as far as possible [27]. Generally, interventions for swallowing difficulty are classified into compensatory and rehabilitative types [7,16]. Rehabilitation is applied for the acceleration of recovery, including surgical therapy and enteral feeding [27]. Yet, 0.4–4.4% of surgeries have been reported to be accompanied by serious complications requiring further intervention [3]. Randomized controlled trials (RCTs) have shown that the use of enteral feeding does not reduce the risk of aspiration and might lead to adverse symptoms such as allergy, nausea, vomiting, bloating, and diarrhea [3,28]. In contrast, most patients still rely on compensatory methods to assist with safe swallowing whether or not they transfer to rehabilitation. Diet modification (or texture-modified foods, TMF), a main compensation strategy, includes foods that are processed by softening, crushing, and pureeing, as well as thickened liquids aimed at the aged with eating dysfunctions [6,7,29,30]. A systematic review involving 26 studies that conducted nutrition intervention through TMF had a positive impact on weight and mealtime satisfaction [31]. Although more high-quality follow-up studies and clinical interventions are needed to make firm conclusions, researchers believe that TMF is a promising strategy for improving nutritional status [19].

## 3. Characteristics, Grades, and Testing Methods of Dysphagia Food

Texture and rheological properties are essential for designing dysphagia foods. Recommendations have indicated that patients with chewing and swallowing difficulties should intake soft, moist, appropriately viscous, and easy-to-swallow foods [5,32]. For solid foods, it is suggested to reduce hardness so that dysphagia patients can swallow food with little or no chewing, as well as consider the water content, food uniformity, and particle size; for liquid foods, the viscosity should be increased to an appropriate level to avoid accidental aspiration. Unfortunately, each country has its own national terminology and grades of liquid and solid food. These guides describe details mostly based on sensory evaluation such as “ thicker than regular liquids”, “similar to the viscosity of pudding or mousse”, “ difficult to absorb with a thick straw”, “it is better to remove with a spoon”, “sticky and pudding-like food”, “do not necessarily require chewing”, and “smooth, uniformly viscous, non-granular, muddy food”, which has been comprehensively reviewed by Yang et al. [33].

In order to reduce clinical confusion and develop standard terminology and definitions to describe food texture and drink thickness for people suffering from dysphagia, as well as to facilitate consistent communication in the field, the framework of the International Dysphagia Diet Standardization Initiative (IDDSI) was announced in 2013 [4] and 6 years later, the completed and objective framework was published. Based on food texture and rheological properties such as hardness, adhesiveness, cohesiveness, particle size, and flow rate, this international system contains a continuum of eight levels (0–7: 0–4 for drinks and 3–7 for foods). Certain solid foods share similar textures with thickened drinks, thus creating an overlap zone in the middle of the framework (levels 3 and 4) [34]. Figure 3A shows detailed definitions of the complete IDDSI framework [35]. In addition, the IDDSI developed different testing methods to classify foods. As shown in Figure 3B, the IDDSI flow test is applied by a 10 mL slip-tip hypodermic syringe to quantify the liquid categories of levels 0–3 according to the remaining sample volume after 10 s of flow [36]. For moderately/extremely thick drinks (level 3/4), further confirmation is required with the supplemental IDDSI fork drip test and/or spoon tilt test. In addition, purees and soft, firm, and solid foods are graded using the IDDSI fork drip test, spoon tilt test, and fork pressure test (Figure 3C).

To date, the IDDSI framework and testing methods have been accepted in hospitals and industry with the practical advantage of being easy to operate and simple to understand. Nevertheless, researchers have suggested that instrumental measurements should be considered in order to perform standardized dysphagia management and quality control [37], such as using rotational shear rheometers to measure the viscosity and viscoelasticity of thickened fluids or applying texture analyzers and tribometers to test the mechanical properties (e.g., harness, adhesiveness, and cohesiveness) and tribology of soft-fish pastes, beef pastes, and chicken stew [32,38]. Meaningfully, Baixauli et al. [39] utilized different texture analyzer settings to discriminate four commercial thickeners through penetration tests, forces, and areas. The results showed that the extrusion tests provided more rheological parameters beyond shear viscosity, such as a cone probe for adhesivity, a disc probe for cohesiveness, and a sphere probe for penetration and elasticity, which suggested that it would make sense to characterize and classify dysphagic food by means of texture analyzers in order to provide safe and effective clinical effects [39].

## 4. Strategies for Food Texture Modification

The physiological dysfunctions arising from the aging process determine that soft, moist, appropriately viscous, and easy-to-swallow foods are preferred. Pureed and minced foods are the simplest technologies to obtain TMF. However, the lack of sensory appeal of these TMF can cause decreased intake and even food refusal [40]. In this section, we will overview three categories of technologies for TMF production that maintain appearance, color, and flavor: (1) gelation to modify the rheology of liquids for safe swallowing [32,38]; (2) conventional nonthermal processes for softening food materials such as enzyme impregnation [40], high-pressure processing (HPP) [5], pulsed electric field (PEF) [41], and ultrasonic (US) methods [42]; and (3) novel 3D printing technology for structural remodeling [32,43].

### 4.1. Application of Thickeners

To achieve swallowing safety, thickening agents are effectively used in clinical dysphagia management to increase the viscosity, smoothness, and cohesiveness of foods and beverages [4]. The mechanism involves the interaction of thickeners with water molecules to form a three-dimensional hydrated network structure and achieve a thickening effect or an increase in viscosity through the entanglement of molecular chains. Such an application can assist dysphagia-based patients in controlling the swallowing muscles to quickly close the respiratory tract and open the food pathway, thereby avoiding aspiration risks. At present, commercially available thickening agents are classified as modified starch-based and gum-based thickeners, some of which are summarized in Table 1 along with their characteristics and applications.

Modified starches prepared through a gelatinization process are developed to produce thickened liquids, mainly including modified corn starch, potato starch [45], tapioca starch [44], taro starch [47], and maltodextrin [46]. For instance, Yang and Lin [44] evaluated the stability of three fluids (distilled water, sport drink, and orange juice) thickened with modified tapioca starch and the results suggested that the employed thickener presented good physical and chemical properties to be potentially utilized for dysphagia-friendly formulations. In addition to these, non-traditional sources of starch such as quinoa starch have been highlighted due to their high freeze–thaw stability and additional nutritional value [56]. Unfortunately, modified starch-based thickeners are not well accepted by patients on account of the limitations of starchy flavor, grainy texture, and cloudy appearance [38]. Furthermore, their viscosity increases with time, easily causing increased residue after swallowing. And they may be hydrolyzed by contact with amylase in saliva, leading to a potential risk of post-swallowing aspiration.

To address these shortcomings, gum-based thickeners with better palatability and viscosity stability are being explored as new alternatives. In particular, xanthan gum is the most commercially used microbial polysaccharide secreted by the bacterium Xanthomonas campestris and its main structure is composed of a β-1,4-D-glucopyranose backbone substituted on alternate glucose residues with a trisaccharide side chain [57]. As a new generation of thickening agents, xanthan gum has desirable properties, including high viscosity with a slippery mouth feel, good hydration properties, time stability, and insensitivity to amylase, temperature, and pH. It is normally used in dysphagia management to prevent complications [49]. Cao et al. [48] investigated the effect of a xanthan-gum-based thickening agent on the incidence of pneumonia and life quality in dysphagia patients aged over 65 years. The results showed that, compared to the control group, the incidence of aspiration pneumonia in the experimental group was significantly lower (*p* < 0.05), while the life quality score was better (*p* < 0.001). Similarly, the finding of Chang et al. [58]’s clinical intervention also indicated that the application of thickening agents could improve swallowing safety. Other hydrocolloids include agar, carrageenan [50], guar gum [4], gellan gum, carboxymethyl cellulose [38], and carboxymethylated curdlan (CMCD). Wei et al. [53] investigated plant- and meat-based diets to provide a good viscous component and a bolus that is easier to swallow, as shown in Table 1. More recently, several new natural thickeners are attracting considerable interest for their additional benefits. Flaxseed gum is a soluble and stable polysaccharide extracted from flaxseed hulls which contains a valuable natural source of phenolic compounds that exhibit pharmacological properties such as anti-diabetic, anti-hypertensive, immunomodulatory, and anti-inflammatory effects and the intake of flaxseed fiber can improve intestinal transport and promote weight control [59]. Wei et al. [53] found that konjac glucomannan (KGM) presented an excellent thickening ability in water and a significant viscosity reduction in model emulsions containing maltodextrin. Given the beneficial characteristics of KGM as dietary fiber, the authors suggested that the antagonistic effect of maltodextrin on its viscosity could provide an opportunity to use more dietary fiber in dietary formulations for dysphagia. Similarly, alginate is an anionic polysaccharide, naturally derived from brown algae, which can undergo ionic cross-linking with calcium ions in an aqueous solution to form an “egg-box model” gel structure [55,60]. Laguna’s [55] research has shown that using such alginate microbeads could delay the entry of food into the pharynx and increase the oral residence time, which would be helpful in designing special foods for people with swallowing disorders. In addition, alginate has been medically found to form a foamy gel on the surface of gastric contents. This barrier-like gel can replace the acid pocket at the esophagogastric junction and protect the esophageal and upper respiratory tract mucosa from acid and non-acid reflux [60]. For this reason, alginate is also frequently applied in gastric reflux therapy and delivery systems for patients with dysphagia [61].

Compared to gum-based thickeners, starch-based thickening agents usually present different rheological properties, such as yield stress and extensional viscosity [57]. Recent studies have shown that starch-based thickeners require greater mass to achieve viscosity similar to that of gum-based thickeners [33]. Moreover, sensory perception is important for the acceptability of thickeners. Some patients do not like the sensory characteristics of starchy thickeners but age and physiological status have been reported to influence the perception of liquid viscosity in the oropharynx [33,62]. Studies have demonstrated that stroke patients with swallowing difficulties expressed a greater taste preference for modified starch-based thickeners [63]. Yang et al. [33] speculated that such thickeners are partially digested by salivary amylase in the mouth, creating a sweet taste and reducing dryness, which in turn makes starch-based thickeners more preferred by older people with dysphagia. Baert et al. [64] also found that broccoli soup prepared with starch-based thickeners exhibited superior taste and aroma intensity. On the other hand, two types of thickeners have different influences on aspiration in dysphagia patients. Xie et al. [65] found that, compared to starch thickeners, xanthan gum thickeners exhibited better effects on decreasing the incidence rates of aspiration and pulmonary injury in dysphagia patients, which was consistent with previous works [33,38]. Nonetheless, partial meta-analyses have demonstrated that there is no convincing evidence that thickened fluids prevent pneumonia caused by dysphagia or that they improve life quality [66]. Accordingly, high-quality clinical trials and long-term observations will be required in future studies. At the same time, the composition, the optimal dose, application temperature, and therapeutic range of each thickener, as well as personalized recommendations, should be considered in clinical treatments.

### 4.2. Nonthermal Processing Technologies

#### 4.2.1. Enzyme Treatment

Enzyme treatment is the impregnation of foodstuff with enzymes which break down cell wall components and/or structures leading to softened textures [29]. Papain and bromelain are commonly used plant proteases for meat tenderization, which is based on the degradation activity on meat connective tissue proteins and myofibrillar proteins of meat [67]. For instance, Ribeiro et al. [68] found that the combination of papain and microbial transglutaminase could be used to develop much softer burgers made from chicken and beef, contributing to oral comfort during the consumption of these products. Similarly, Eom et al. [69] utilized a 1.0% concentration of bromelain and collupulin to soften the chicken breast and eye of round beef, providing food alternatives for elderly individuals who have problems with mastication. It is also interesting to note that texture-modified pumpkin [70] and softened kombu [71] have been developed by using enzyme impregnation. These products retained their original appearances while being soft enough to be completely mashed with the tongue and upper jaw, making them ideal for elderly people. Furthermore, in order to improve this softening effect and its efficiency, enzyme impregnation is often used with other techniques. Botinestean et al. [72] optimized an innovative beef product formulation method to achieve reduced chewiness through 120 min sous-vide cooking and adding papain (0.01 mg/100 g). Based on freeze–thaw enzyme infusion, senior individuals with poor masticating function can easily enjoy hard and fiber-rich vegetables such as lotus rhizome and carrot [73].

#### 4.2.2. High-Pressure Processing

High-pressure processing (HPP) is a non-thermal technique by which food is treated at a pressure of more than 100 MPa through a compressing fluid [74,75]. Due to the ability to improve the gelation behavior of protein by changing its conformational structure, HPP has been noted for creating innovative foods, especially meat and meat products. Figure 4A displays a scheme of changes in the protein structure. Without the application of high pressure, two or more folded peptide chains or subunits are tightly arranged to form a quaternary structure, which is maintained by hydrophobic interactions [5]. Furthermore, HPP treatment with 100 MPa–200 MPa disrupts the non-covalent hydrophobic and electrostatic interactions and hydrogen bonding between protein molecules, which results in the stretching and unfolding of the quaternary structure of the protein [76]. At a pressure higher than 200 MPa, the tertiary structure would suffer conformational modifications: hydrophobic groups previously buried in the internal regions of the protein and surrounded by a non-polar environment would be exposed to an aqueous environment, leading to the dissociation into an unstable secondary structure that is present as alpha helices or beta sheets [5,76]. Such changes also offer the opportunity to lead to protein denaturation, cohesion, aggregation, and gelation [74,77]. Furthermore, a pressure over 700 MPa irreversibly disrupts the secondary structure into the primary structure, which is not damaged by HPP, as the amino acid chains are connected by covalent bonds that hardly rupture [5,78]. Zhang et al. [76] manifested a lower hardness of chicken gels with increasingly high pressure from 100 to 500 MPa and also suggested that HPP treatment at ≥ 300 MPa could be more suitable for the production of meat-based dysphagia foods. As reported by Tokifuji et al. [79], pork meat gel prepared at a meat/water (*w*/*w*) ratio of 1:1 and treated with 400 MPa for 20 min at 17 ± 2 °C showed good smoothness, softness, and elasticity. In addition, a VFSS test also revealed that this gel left little residue in the oropharynx, indicating that HPP treatment could be used to create dishes for a dysphagia diet. Three years later, the same technology and processing parameters were used to prepare fish meat gels with a meat/water (*w*/*w*) ratios of 1:1 and 1:1.5 that conformed to the Japanese Dysphagia Diet 2013 criteria [80]. Additionally, several researchers have applied HPP technology to the production of starch-based and fruit-based products with low hardness, springiness, and chewiness, such as buckwheat starch [81] and apple purée [82], which could be considered adequate TMFs for people with chewing and swallowing difficulties.

#### 4.2.3. Pulsed Electric Field

A pulsed electric field (PEF), a non-thermal technique, is performed by the application of short and intense electric field pulses to a food material located between two or more electrodes [41,83]. In the presence of such short-duration external pulses, the cell membranes of food ingredients are electroporated, which in turn leads to local structural changes. Figure 4B illustrates the mechanism of PEF treatment. Depending on electric field parameters such as intensity, frequency, treatment time, pulse length and shape, and food material properties, PEF can lead to reversible or irreversible electroporation of the cell membrane. Specifically, reversible electroporation means that the formation of small pores is transient and that the cell membrane is resealed after the external field’s impact has ceased, which can be used to incorporate different bioactive substances into biological cells [84,85]. However, when irreversible electroporation occurs, permanent membrane disruption leads to cell breakdown and the leakage of intercellular contents; this is mainly used for freezing, drying, osmotic dehydration, extraction, or microbial inactivation [41]. Such nonthermal technology has a positive effect on food color and even taste, so it has been successfully used for meat and plant processing. Huo et al. [86] explored the effects of different pulse electric field intensities (1.00, 2.25, and 3.50 kV/cm) on the palatable quality of beef. The results showed that, at 3.50 kV/cm, the hardness, cohesiveness, chewiness, and elasticity of beef were significantly reduced compared to the control group. This is similar to the results of Jeong et al. [87], which suggested that the shear force, chewiness, and hardness of beef decreased as the electric field intensity increased. Such a phenomenon may rely on the fact that PEF treatment could increase the degradation of troponin and pro-myosin in beef, resulting in improved tenderness. Moreover, because PEF technology not only retains heat-sensitive components such as vitamins, carotenoids, and phenols but also increases the extractability of bioactive compounds, it has been applied to soften apple, potatoes, carrots [88], and sweet potato tubers [89]. Jin et al. [90] revealed that PEF treatment caused a softer texture of blueberries but hardly changed the color or appearance. And after PEF treatment, the anthocyanins and phenolic contents increased by 10% and 25%, respectively.

#### 4.2.4. Ultrasonic

Ultrasonic (US) is a type of sound wave with a frequency higher than can be heard by humans (≥20 kHz) [41]. It is considered as another effective technology that can be applied to improve food texture and quality due to its ability to penetrate various substances. Depending on the frequency, the US employed in the food industry can be divided into low-frequency US (20–100 kHz) and high-frequency US (1–10 MHz) [91,92]. Among these, high-frequency US is commonly used for non-destructive testing, process control, and quality evaluation as well as for monitoring food packaging materials, as it does not modify the physical or chemical properties of the material [41,92], while low-frequency US is useful for generating cavitation through high power levels and, thus, is applied in the food processing processes of drying, freezing, extraction, emulsification, homogenization, enzyme inactivation, and sterilization [41,91]. Dong et al. [42] and Alarcon-Rojo et al. [91] reviewed various studies elucidating the effects of different meat and meat products in detail, which might lay a foundation for the production of dysphagia-based foods. For example, Wang et al. [93] found that beef semitendinosus showed an increased myofibrillar fragmentation index (MFI) value and the proteolysis of desmin and troponin-T after US treatment with 20 kHz frequency for 20 or 40 min at 3 and 7 days of post-mortem aging, indicating an improvement in beef tenderness. The mechanism of US for tenderizing meat relies on the cavitation effect, which can be interpreted as a series of non-linear processes including the formation, expansion, compression, and collapse of tiny bubbles or cavities over a short period of time, caused by the propagation of sound waves between positive and negative pressures [42], which is shown in Figure 4C. At the time of collapse, the bubbles release high amounts of energy, triggering extreme pressure (50–100 MPa) and temperature (5500 °C) conditions as well as generating some physical effects such as microstreaming, jetting, free radicals, and high shear stress, which could cause cell membrane damage and changes in protein properties. Such effects may contribute to the depolymerization of actin filaments or the breakdown of myogenic fibrin, which in turn increases the meat tenderness [41,94]. For starch-based foods and vegetables, US treatment modifies chewiness mainly by affecting the viscosity of the materials. Sit et al. [95] demonstrated that US treatment with a 30 kHz frequency for 5 min resulted in the highest solubility, swelling, and viscosity of taro starch. Anese et al. [96]’s work revealed that the viscosity and lycopene release of tomato pulp increased with US treatment of 24 kHz frequency for 30 min.

#### 4.2.5. Overview of Mentioned Technologies

It can be seen that the applications of enzyme, HPP, PEF, and US have effects on the textural and physicochemical characteristics of meat, fish, carbohydrate-based products, and fruit and vegetables, which provides the possibility to produce dysphagia-oriented foods (as summarized in Table 2). The advantages of such technologies include negligible loss of vitamins, nutrients, volatile compounds, and flavor, as well as high convenience and non-destructiveness [91]. Nevertheless, it is too early to judge which technology would produce the best results in terms of texture modification. The textures of softened diets depend not only on the parameters of these techniques but also on the type and size of raw materials [5]. It has been reported that free radical chain reactions triggered by the US cavitation effect could promote the oxidation of fat and protein, contributing to unpleasant flavors [42]. HPP tends to change the color of meat, which is one of the most important quality features to consider when consuming. Furthermore, different patients with swallowing disorders may present distinct symptoms and taste preferences, creating different texture requirements. Consequently, more considerations are necessary for selecting suitable technology and imaging techniques (such as VFSS) and should be implemented to assess the suitability and swallowability of foods prepared by these techniques.

### 4.3. Designing Food Texture with 3D Printing

Three-dimensional printing, known as additive manufacturing, is a layer-by-layer construction technology in three-dimensional spaces [97,98]. Specifically, it refers to the process by which material layers are joined or solidified to shape 3D objects in geometrically complex shapes under the control of computer software. Due to its advantages of high precision, fast speed, and low cost, 3D printing technology has been considered to have potential for industrial revolution and has been greatly applied in the areas of biological organs, healthcare, manufacturing, and construction [99]. In 2007, this technology was used for the first time by researchers at Cornell University in the food sector to allow individuals to print out chocolate at home [100,101]. Since then, the application of 3D printing technology for food fabrication has stimulated the interest of manufacturers and academia for its potential to customize food products according to necessities and preferences.

Currently, 3D printing techniques available in food production mainly include extrusion-based printing (or fused deposition modelling), selective sintering printing, binder jetting, and inkjet printing [102,103,104,105]. Among them, extrusion-based printing is the most widely used in the food sector due to the multiple choices of printed materials and simple operation of the printing system [106]. With this technology, a semi-solid or fluid material is extruded from the nozzle by power from a syringe, the air, or a screw, then solidified and deposited layer by layer onto the printer bed [107]. Figure 5 displays the pictorial flow of the 3D food printing process based on extrusion technology. Indeed, food printing starts with the pretreatment of food materials and the design of formulations, as the printing properties of ingredients need to be taken into account [101]. The next stage consists of creating 3D models through computer-aided design (CAD) (https://www.autodesk.com.cn/ (accessed on 4 January 2024)) or scanning an existing object. Then, the model is transferred to slicing software, where the model layers are created and parameters such as the layer height, nozzle speed, extrusion rate, and temperature are set [105]. After that, a G-code is sent to the printer, which then fabricates the pre-defined structure. Finally, the product is treated through freezing, baking, or frying to make it edible [43,101,107,108].

As mentioned above, the characteristic of food materials required for extrusion-based printing is viscous but not too thin, which creates great possibilities for the creation of 3D-printed soft foods to meet the requirements of the elderly and patients with poor swallowing abilities [106]. For instance, dysphagia-friendly pork and beef products have been successfully printed using different hydrocolloid additions. For pork paste, samples containing xanthan gum and guar gum, either independently (0.36%) or combined (0.36% in proportion 0.5:0.5, 0.7:0.3, and 0.3:0.7), presented excellent rheology and extrusion-based printability and thus were categorized as potential transitional foods in the IDDSI framework (levels 5–7) [109]. During the process of printing beef paste, the individual or mixed addition of cold-swelling (xanthan and guar gum) and heat-soluble (κ-carrageenan and locust bean gum) hydrocolloids were analyzed and suitable products with different formulations were classified within IDDSI levels 5–7 through texture profile analysis (TPA) and IDDSI testing methods [110]. Similarly, other 3D-printed meat products for people with swallowing difficulties also include pureed chicken and tuna [111]. When it comes to vegetable/fruit-based foods, Pant et al. [112] selected three distinct fresh vegetables (garden peas, carrots, and bok choy), along with hydrocolloids (xanthan gum or/and locust bean gum), to print nutritious and tasty foods. Also, printed products were assessed for their suitability for dysphagia patients by the IDDSI spoon tilt and fork pressure tests. Qiu et al. [113] used apple and edible rose as raw materials, with the addition of xanthan and basil seed gum, to produce level 5 minced and moist dysphagia foods. Black fungus had health benefits but required certain chewing efforts [114] and Hypsizygus marmoreus by-products with medicinal values [115] were also developed for attractive 3D-printed dysphagia diets with the incorporation of xanthan gum. Lee et al. [116] used egg white and alternative food ingredients to study the foods’ foaming properties, rheological properties, and printability for special diets through printing food foams, suggesting the potential to improve textural properties using xanthan gum. Recently, Zhang’s research team employed a similar approach to construct two traditional Chinese foods for the elderly with dysphagia, including mooncake [117] and Qingtuan [118]. Through the texture and IDDSI tests, the best formulations were chosen and the final mooncake and Qingtuan products were categorized as IDDSI levels 4 and 6, respectively. Table 3 lists detailed examples of the production of dysphagia foods using 3D printing technology.

In summary, the 3D printing technique has attracted a significant amount of attention in food production for its advantage of freedom to design novel structures and textures and to develop personalized nutrition and flavors [119,120]. Also, mechanized production simplifies and speeds up the manufacturing process, in addition to reducing food waste [101,105]. In spite of this, 3D food printing for dysphagia has the following disadvantages: high cost, limited compatible materials, slow mass printing, and unattractive appearances. Furthermore, a considerable number of consumers have shown concern or even negative attitudes towards the food safety and nutritional values of 3D-printed products [119,121]. Consequently, it is hoped that more alternative ingredients with environmental, nutritional, and economic values, such as surpluses and by-products from food production or novel protein sources from insects and algae, can be developed to facilitate the application of this technique. Additionally, the development of nutritional fortification technologies, such as electrostatic spinning and microencapsulation, in the 3D printing sector should be considered to meet customized demands for healthier foods [122]. And hopefully, 3D food printing can be combined with artificial intelligence methods for the purposes of searching for potential materials, optimizing the printing formulation, and predicting the printing fidelity by means of machine learning and genetic algorithms or artificial neural networks [43]. However, there is a long way to go in terms of raising consumers’ perception, for which safety assessments and extensive advertisements are necessary.

## 5. Nutrition and Sensory Improvement in TMF

Owing to physiological dysfunctions or/and aging-related diseases, impairments in eating ability and sensory perception are common in the older, which causes reduced food consumption, thereby leading to deficiencies in energy and nutrients and even malnutrition. As a result, nutritional fortification and flavor enhancement are highlighted as ways to accommodate specific needs and alleviate malnutrition in the aged population.

On the one hand, the elderly have an increased demand for nutrients such as protein; amino acids; many micronutrients, especially vitamins B, D, C, and E; calcium; and dietary fiber that are linked to body weight, muscle mass, and gut health [15,123]. Garcia et al. [124] designed a functional food on the basis of the protein hydrolysate from sea cucumber to specifically meet the nutritional needs of the western elderly. Giura et al. [125] provided nutritional vegetable purees with significant amounts of proteins (6.1–6.7%) and antioxidant bioactive compounds (43–53 mg phenolic compounds/100 g) for dysphagic people. Kersiene et al. [126] prepared a stable double emulsion with several bioactive substances (adding vitamins B6, B12, and C and anthocyanins in the aqueous phase and vitamins A and D3 in the oil phase) for formulations tailored to the elderly, which showed that the encapsulated compounds were stable and such active substances could be released during digestion. Xie et al. [120] considered that 3D food printing is crucial to meet requirements for the elderly regarding energy supplementation (protein enrichment and fat reduction), flavor regulation, and nutrition balance. Their review demonstrated that the addition of whole eggs, whey protein, oats, and peanuts in 3D-printed food increased the nutritional profiles. And 3D-printed emulsion gels or oleogels could include bioactive substances in order to promote the health of elderly people. Furthermore, Costa et al. [127]’s dietary study suggested that dysphagia patients without nutritional risks should consume 1750 kcal of energy (25 kcal/kg/day), 70 g protein, and 1750 mL water/day, while 2037 kcal of energy (35 kcal/kg/day), 90 g protein, and 2000 mL water/day need to be provided for malnourished patients. In the market, IFF Nourish^®^ has launched a thickening agent compounded by xanthan gum and guar gum. The solution also considers the addition of protein, fiber, and probiotics to improve nutritional deficiencies in elderly patients with swallowing difficulties. Maruha Nichiro Corporation (https://www.maruha-nichiro.com/ (accessed on 2 May 2023)) in Japan developed a series of products called “Protein 21”, which contain more than 21 g of protein per 100 g of ingredients, specifically for the elderly, meeting the requirements for a low portion but high nutrition.

On the other hand, it has been reported that the elderly present a greater preference for taste-fortified foods [14,15]. Natural ingredients with intense flavors, such as soy sauce, garlic, onions, ginger, basil, and leeks, increased the average energy intake among elderly hospitalized patients by 13–26% [128]. A recent study formulated different protein beverages with the same viscosity for elderly people with dysphagia, showing that the beverages flavored with meat broth received higher comfort and ease scores compared to the beverages flavored with mushroom [129]. Other methods, such as microencapsulation of effervescent powders and tiny high-pressure bubbles of carbon dioxide, were also applied to change the flavor of pureed food by promoting the activation of the trigeminal pathways [14,130]. It is worth noting that there is significant heterogeneity in taste and smell impairment, which prevents flavor enhancement from being widely applied. Ribes et al. [131] suggested that packet sauces might be an alternative way, since the elderly could choose according their preferences and add them as needed.

## 6. Current Aged Food Status Based on Dysphagia

In general, meals for older adults can be divided into regular and soft foods. Elderly people with normal chewing abilities can obtain regular meals through home cooking or institutional supply, while soft foods are more preferred for people with chewing and swallowing dysfunction or for the oldest of the elderly. Japan has been an aging society since the 1970s, which has ignited the idea of developing “care food”. In 2002, the Japan Care Food Association was established and issued the universal design food (UDF) standard, which involves adjusting the shape, texture, and other aspects of processed foods according to the eating ability of the consumer. Since then, several initiatives to guide diets for the elderly were issued in Japan, including the Dysphagia Diet in 2013 and the “Smile Care” food in 2015. “Smile Care” food is categorized using different color labels, with blue for normal individuals without chewing or swallowing problems, yellow for those with chewing disorders, and red for those with swallowing disorders. To date, approximately 70 companies in Japan have developed aged-care foods, such as Maruha Nichiro, Kewpie, Meiji, Morinaga, etc. (https://www.udf.jp/outline/members.html (accessed on 2 May 2023)), driving the market’s rapid growth for 10 consecutive years. The total domestic market of nursing, elderly and therapeutic foods reached at JPY 173.6 billion in the fiscal year of 2019 in terms of the manufacturer’s shipments [14]. Furthermore, certain food companies in South Korea have exploited various soft foods through enzyme technology, such as rice cakes, soft fish, grilled meat, and even burger steak with insect powder, which are especially suitable for elderly people to intake and digest [14]. Hormel Health Labs, established in America, provides a full range of safe, affordable, and tasty aged foods for the population with dysphagia, including Thick & Easy^®^ pureed foods, dessert purees, and tea sticks (https://www.hormelhealthlabs.com/products/condition/dysphagia-swallowing difficulties/ (accessed on 2 May 2023)). Nutricia (Fulda, Germany) and Nestlé Portugal S.A. (Linda-a-Velha, Portugal) also commercially developed starch and gum thickeners for dysphagia-oriented clear liquids such as Nutilis^®^ and Boost^®^ Nutritional Pudding [40].

However, compared with the sound aged food industry in foreign countries, this market in China is relatively lacking. At present, there are few varieties of food specially exploited for the elderly, most of which are cakes, biscuits, paste-type foods (oatmeal paste and sesame paste), and prepared milk powder with good palatability for swallowing. However, such foods cannot meet the daily nutritional supplement needs of senior citizens and the changes in sensory level among the aged is less considered in food design. Kang et al. [132] used a cross-sectional survey to collect and analyze the nutritional claims of 220 prepackaged aged foods in China. The results showed that the types and contents of nutrients added to the powdered dairy products varied and that certain products claiming to be “nutritional formulas” could not meet the physiological requirements of the elderly. In addition, although pastry products can be chewed and digested easily, the contents of fat, sugar, and salt are commonly high, which is not suitable for long-term consumption by the elderly. Consequently, it is necessary to strengthen the research on geriatric food and nutrition in order to meet the diversified health needs of Chinese elderly people at different levels.

## 7. Conclusions and Future Prospective

Dysphagia is a common problem with a higher prevalence in the elderly. People with swallowing dysfunction are more susceptible to suffering from malnutrition, dehydration, aspiration pneumonia, or even death. Therefore, developing soft, moist, and palatable texture-modified diets is pressingly needed. This review clarifies changes in the physiology of oral processing and swallowing in the elderly. According to the characteristics of dysphagia foods, traditional non-thermal technologies (enzyme treatment, HPP, PEF, and US), gelation, and 3D printing for chewing and swallowing safety in older individuals are summarized herein. Meanwhile, flavor enhancement and nutrition enrichment are considered to compensate for the loss of sensory and nutrients.

It should be pointed out that there are still multiple challenges in exploring special foods for the elderly with dysphagia. Firstly, the IDDSI framework has provided feasible testing methods and grading criteria for dysphagia-oriented food texture but it is easily affected by subjective factors. Additional rheological parameters, such as the viscosity, viscoelasticity, hardness, cohesiveness, adhesiveness, and tribology of food boluses during swallowing, remain to be investigated, which will constitute evidence for quality criteria and clinical guidelines. Next, it is noticeable that there are limited and disjointed comprehensive research studies on the long-term impacts of thickened fluids and TMF on mastication and digestion in the elderly to some extent. Bolus formation and disintegration determine the rate of nutrient absorption and release into the body. In this regard, an in vitro dynamic simulation system particularly targeting elderly individuals will become meaningful and valuable and it may shed more light on the design of aged foods. Thirdly, we should attach importance to elderly individuals with co-morbid conditions such as hypertension, diabetes, and sarcopenia, who have special dietary demands such as low salt, low sugar, or high protein. Designing such foods requires an integrated consideration of nutrition, taste, texture, and medical effect. Undoubtedly, the advance of aged foods requires strengthened multidisciplinary cooperation with nutrition, food technology, geriatrics, and healthcare and is essential to tackling complex challenges associated with designing and developing suitable diets for the elderly with swallowing disorders. Addressing the above limitations will provide a more complete understanding of the dietary requirements of older adults with dysphagia and will contribute to improved solutions for healthy aging.

## Figures and Tables

**Figure 1 foods-13-00215-f001:**
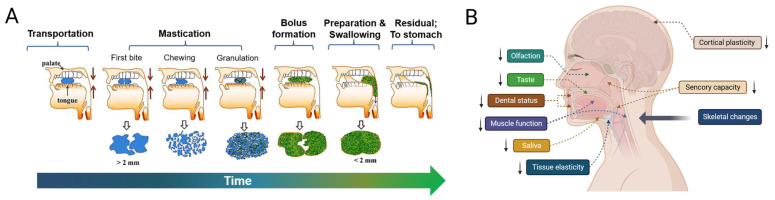
(**A**) The whole process of oral food processing and swallowing. (**B**) Several important risk factors are associated with dysphagia in the older population. Notably, ↓ indicates decreased function.

**Figure 2 foods-13-00215-f002:**
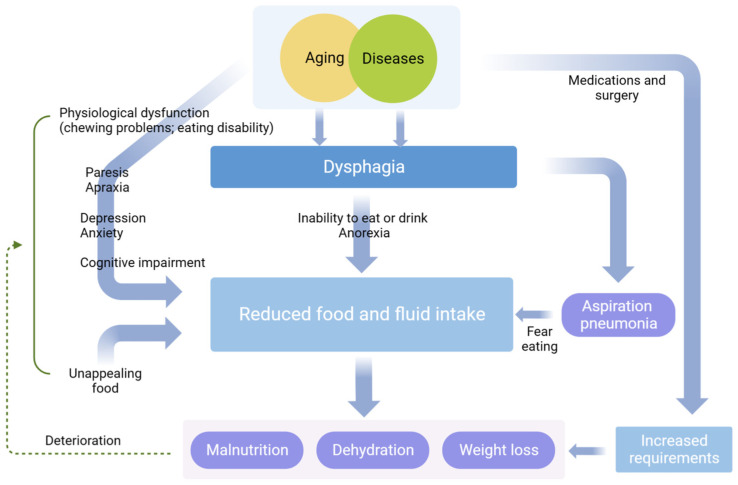
The impacts of dysphagia on the health and life quality of older adults.

**Figure 3 foods-13-00215-f003:**
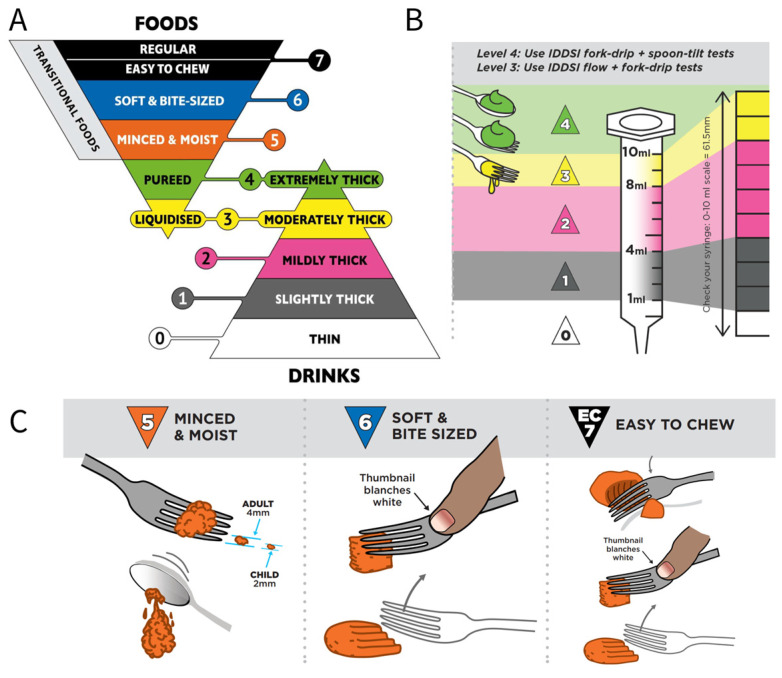
(**A**) The complete IDDSI framework’s detailed definitions of 2019, used for individuals with dysphagia, including texture-modified foods and thickened liquids (https://iddsi.org/Framework (accessed on 18 April 2023)). (**B**) Flow test for levels 0–4 with the IDDSI framework (https://iddsi.org/Testing-Methods (accessed on 18 April 2023)). (**C**) Food test for levels 5–7 with IDDSI framework (https://iddsi.org/Testing-Methods (accessed on 18 April 2023)).

**Figure 4 foods-13-00215-f004:**
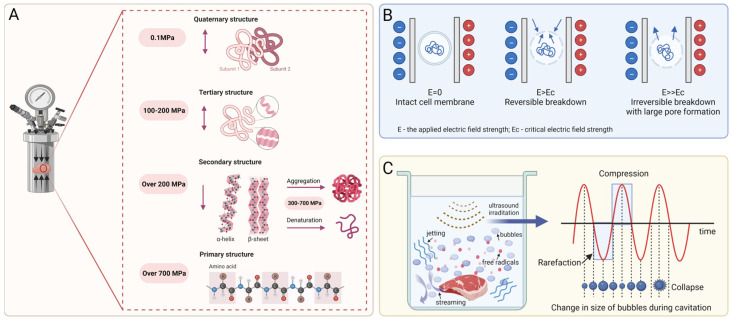
(**A**) Structural changes in proteins under high-pressure processing treatment. The bidirectional arrow indicates that the protein structure is reversibly changed during the process, while the unidirectional arrow indicates that the structural change is irreversible. (**B**) Schematic representation of the pulsed electric field mechanism. The arrow indicates reversible or irreversible perforation of the cell membrane. (**C**) Diagrammatic sketch of the impact of ultrasound on meat and ultrasonic cavitation effect. The red wavy line indicates a series of non-linear changes of acoustic pressure with time.

**Figure 5 foods-13-00215-f005:**
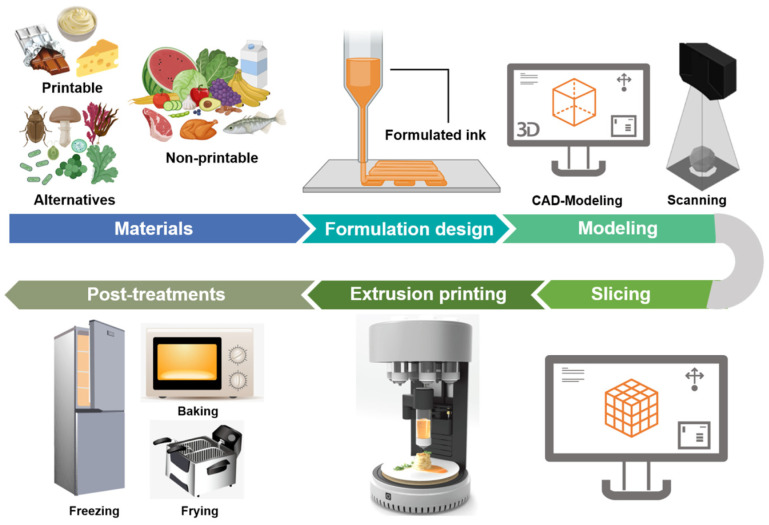
General stages of 3D food printing process based on extrusion technology.

**Table 1 foods-13-00215-t001:** Some modified starch-based and gum-based thickeners applied in dysphagia management.

Type	Characteristics	Thickeners	Application	Results	References
Modified starch-based thickeners	Low-cost Environmental sensitivity Starchy flavor Grainy texture Cloudy appearance Increased prevalence of pharyngeal residue	Tapioca starch	Distilled water Sport drink Orange juice	All thickened fluids with a nectar-like consistency (300 ± 20 mPa·s) appeared to be shear-thinning fluids with yield stress closely fitting the power law and Casson models.	[44]
		Potato starch	Water Juice Milk	The milk samples showed the highest viscosities, with the consistency of “pudding” (>1750 cPs), and milk with potato starch achieved good acceptability.	[45]
		Corn starch		The fluids with corn starch showed the consistency of “honey” (351–1750 cPs).	
		Maltodextrin	Water Orange juice Tea	The powder prepared by maltodextrin and soy protein increased beverage viscosity with a nectar-like consistency (51–350 cPs).	[46]
		Taro starch	Paste	Taro paste was found to have greater softness, less stickiness, and better cohesiveness.	[47]
Gum-based thickeners	Good hydration properties Environmental stability Clear appearance Tasteless and odorless Low risk of aspiration	Xanthan gum	Fruits Vegetables Water Milk Pork	Xanthan gum showed good shear-thinning behaviors, stable properties, excellent palatability, and a suitable texture for dysphagia-friendly formulations.	[48,49]
		Agar	Pureed banana	Agar showed low elastic behavior and was within the safe-swallowable range.	[50]
		Carrageenan	Pureed carrots	Carrageenan had a brittle texture; hence, the food was difficult to swallow.	[51]
		Carboxymethyl cellulose	Thickened pea cream	It provided a greater viscous component and the bolus was easy to swallow.	[52]
		Tara gum			
		Konjac glucomannan (KGM)	Water Model emulsions	KGM showed excellent thickening capacity in water, while a significant viscosity reduction was observed in model emulsions containing maltodextrin.	[53]
		Carboxymethylated curdlan (CMCD)	Water Model emulsions	CMCD in both water and emulsions showed high viscosity, strong shearing behavior, and appropriate viscoelasticity.	[53]
		Flaxseed gum	Water Orange-flavored soy juice Skim milk	Flaxseed gum reduced the coefficient of friction in water and milk but increased the value in soya juice at higher concentrations.	[54]
		Gellan gum	Pureed carrots	A significantly lower concentration of gellan gum could present similar properties with xanthan gum.	[51]
		Guar gum	Water Apple juice Low-fat milk	Enhanced viscoelastic properties were observed when guar gum was dissolved in milk but a reduced extensional viscosity was found in apple juice.	[49]
		Alginate	Distilled water	The incorporation of alginate beads achieved the sensory effect of delaying food entrance into the pharynx and increasing oral residence time.	[55]

cPs: centipoise; mPa·s: millipascal second; both cPs and mPa·s are units of viscosity, 1 cPs = 1 mPa·s.

**Table 2 foods-13-00215-t002:** Traditional processing technologies used in texture-modified foods for the elderly.

Technology	Applications	Processing Conditions	Results	References
Enzyme treatment	Beef and chicken burgers	0.2% papain and/or 1% microbial transglutaminase for 4 h at 5 °C	Papain resulted in a pronounced increase in the softness of beef and chicken.	[68]
	Chicken breast and eye of round beef	1.0% bromelain for 24 h at 4 °C	The hardness of bromelain-treated chicken breast and eye of round beef reached 1.4 × 10^4^ N/m^2^ and 3.2 × 10^4^ N/m^2^, respectively, while the shapes did not change.	[69]
	Seaweed kombu	1 g/100 g protease with 0.3 mol/L sodium phosphate buffer (pH 8.0) for 15 h at 4 °C	Phosphate buffer resulted in a softened texture and proteolysis resulted in reduced stickiness.	[71]
	Pumpkins	Vacuum enzyme impregnation for 1 h at 36 °C	The pumpkins were softened and the treatment had positive effects on antioxidant capacity.	[70]
	Beef meat	0.01 mg/100 g papain with 120 min sous-vide cooking	Good parameters, such as shear force values, chewiness, TPA hardness, cooking loss, and color, were received.	[72]
	Lotus rhizome and carrot	0.25% cellulase solution with different vacuum pressures (0–0.05 MPa), vacuum times (5–30 min), and restoration times (0–120 min)	This freeze–thaw treatment can soften foodstuffs but retain their original shapes.	[73]
HPP	Pork meat gel	Meat/water (*w*/*w*) ratio of 1:1 with 400 MPa for 20 min at 17 ± 2 °C	Good smoothness, softness, and elasticity, and little residue was observed.	[79]
	Fish meat gel	Meat/water (*w*/*w*) ratio of 1:1 and 1:1.5 with 400 MPa for 20 min at 17 ± 2 °C	The food conformed to the Japanese Dysphagia Diet 2013 criteria.	[80]
	Buckwheat starch	120 to 600 MPa for 20 min	With pressure, the hardness, swelling, and viscosity decreased.	[81]
	Apple purée	0 to 600 MPa at 20 °C for 5 min	At 400 MPa, the active compounds were unaffected but the viscosity increased.	[82]
	Beef	1.00, 2.25, 3.50 kV/cm	The condition of 3.50 kV/cm resulted in a reduced hardness of beef meat.	[86]
PEF	Sweet potato tubers	0.3 to 1.2 kV/cm, 540 pulses of 20 μs pulse width, and 50 Hz	PEF treatment reduced the energy required for cutting.	[89]
	Blueberries	2 kV/cm electric field strength, 1μs pulse width, and 100 pulses per second for 2, 4, and 6 min	PEF treatment caused a softer texture and an increased level of anthocyanins and phenolics.	[90]
US	Beef	20 kHz frequency for 20 or 40 min	An improvement in beef tenderness was observed.	[93]
	Taro starch	30 kHz frequency for 5 min	The highest solubility, swelling, and viscosity of taro starch were observed.	[95]
	Tomato pulp	24 kHz frequency for 30 min	An increased viscosity and lycopene release of tomato pulp was tested.	[96]

**Table 3 foods-13-00215-t003:** Three-dimensional printing technologies used in dysphagia management.

Food Applications	Products	Physical Properties	Classification by IDDSI Tests	References
Pork and beef pastes	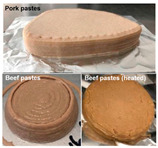	Samples with hydrocolloids presented lower hardness, cohesiveness, and chewiness	Level 5–7	[109,110]
Garden peas, carrots, and bok choy	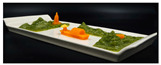	Adding very low amounts of hydrocolloids led to great 3D printability and food palatability.	Level 4	[112]
Apple and rose		Combining xanthan gum and basil seed gum displayed higher gumminess, stiffness, and self-supporting ability.	Level 5	[113]
Black fungus	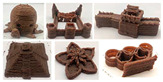	The addition of xanthan gum displayed good self-supporting capability and smooth surface texture.	Level 5	[114]
Hypsizygus marmoreus by-products	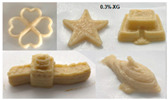	Incorporating xanthan gum and locust bean gum showed better 3D printability, mouth feel, and swallowing easiness.	Level 5	[115]
Mooncake and Qingtuan	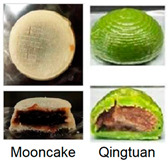	Adding soybean oil and Arabic gum reduced the hardness and adhesiveness of mooncake. Using soluble soybean polysaccharide decreased the hardness and adhesiveness of Qingtuan.	Level 4 (mooncake) Level 6 (Qingtuan).	[117,118]
Egg white food foam	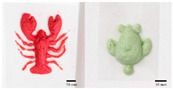	Utilizing egg white and xanthan gum led to minimal water seepage and excellent foam stability.	Level 4–6	[116]

## Data Availability

Not applicable.

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
