# Peer review of "Food Processing and Nutrition Strategies for Improving the Health of Elderly People with Dysphagia: A Review of Recent Developments"

_foods, 2024, doi:10.3390/foods13020215_

Round 1

Reviewer 1 Report

Comments and Suggestions for Authors

Ting Liu et al. presented a good and comprehensive article about Food processing and nutrition strategies for improving the health of elderly people with dysphagia.

1- Please specify your search strategy in this article. State exactly which articles were selected and in what time frame they were studied and processed.

2.  For the convenience of the readers, make a list of all the abbreviations in the article and place them at the beginning of the article.

Reviewer 2 Report

Comments and Suggestions for Authors

The review titled “Food processing and nutrition strategies for improving the health of elderly people with dysphagia: A Review of recent developments” is quite interesting. The manuscripts deal from which are the physiological problems that cause dysphagia. Additionally, characteristics, grades and testing methods of dysphagia foods are treated and some food texture modifications with thickeners are recomplicated. Finally, nonthermal technologies that can be useful for texturizing food such as enzyme treatments, high-pressure processing, pulsed electric field and ultrasonic treatment are summarised. So it is a quite complete document in which the last developments in food treatment for dysphagia patients are resumed.

I think that this review collects last knowledge in dysphagia food treatments and for that reason, it has important relevance in food technology and to dysphagia nutritional professionals.

My only concerns are related to:

In line 195, maybe it will be interesting to include some more information about different texture analyser settings for the characterization of the gels. A nice piece of information can be found in this manuscript.

Baixauli, R.; Bolivar-Prados, M.; Ismael-Mohammed, K.; Clavé, P.; Tárrega, A.; Laguna, L. Characterization of Dysphagia Thickeners Using Texture Analysis—What Information Can Be Useful? Gels 20228, 430. https://doi.org/10.3390/gels8070430

Line 205 I missed the mention of alginate as a thickener agent. I recommend including some information about it in the application of thickeners section.

Ciprandi G, Damiani V, Passali FM, Crisanti A, Motta G, Passali D. Magnesium alginate in patients with laryngopharyngeal reflux. J Biol Regul Homeost Agents. 2021 Jan-Feb;35(1 Suppl. 2):61-64. doi: 10.23812/21-1supp2-12. PMID: 33982541.

Reviewer 3 Report

Comments and Suggestions for Authors

Collecting information in tables helps illustrate the topic. The presented figures, which are very comprehensive, make a very good impression.

It is important to analyze the grammar structure and punctuation and to state clearly the implications for research, practice, and society. The structure and clarity of the manuscript could be improved.

The manuscript does not provide specific data on the prevalence of dysphagia in the elderly population or the current effectiveness of existing therapies. A more detailed presentation of these aspects could strengthen the arguments.

While risks associated with dysphagia are mentioned, there are no specific details on how these risks impact patients' health and how they can be avoided or treated.

The manuscript mentions "nonthermal food technologies, gelation, and 3D printing." Still, it does not provide sufficient information on how these technologies are implemented or the research studies supporting their effectiveness in dysphagia treatment.

There are some coherence and clarity issues in sentence structure, which can make the manuscript challenging for readers unfamiliar with the medical field.

Although IDDSI provides a general framework for classifying food textures, there is a lack of detailed rheological parameters such as viscosity, viscoelasticity, hardness, cohesiveness, and adhesiveness, which could provide a better understanding of the behavior of foods during chewing and swallowing.

There is a lack of comprehensive and coordinated research on the long-term impact of using thickened liquids and modified texture diets on the processes of chewing and digestion. Bolus formation and disintegration, which determine the rate of nutrient absorption and release into the body, remain insufficiently explored.

Older individuals with co-morbid conditions such as hypertension, diabetes, and sarcopenia may have specific dietary requirements, including low salt, low sugar, or high protein. Designing foods that meet these diverse needs requires a multidisciplinary approach involving nutrition, food technology, geriatrics, and healthcare.

The advancement of aged foods requires collaboration among various disciplines, including nutrition, food technology, geriatrics, and healthcare. Strengthening multidisciplinary cooperation is essential for addressing the complex challenges associated with developing suitable foods for the elderly with dysphagia.

Addressing these limitations will lead to improved solutions and a more comprehensive understanding of the dietary needs of older individuals with swallowing difficulties.

Comments on the Quality of English Language

It is important to analyze the grammar structure and punctuation.

Round 2

Reviewer 3 Report

Comments and Suggestions for Authors

The manuscript has been significantly improved, in agreement with the reviewers’ recommendations.